# ChatGPT’s Skills in Statistical Analysis Using the Example of Allergology: Do We Have Reason for Concern?

**DOI:** 10.3390/healthcare11182554

**Published:** 2023-09-15

**Authors:** Michal Ordak

**Affiliations:** Department of Pharmacotherapy and Pharmaceutical Care, Faculty of Pharmacy, Medical University of Warsaw, Banacha 1 Str., 02-097 Warsaw, Poland; michal.ordak@wum.edu.pl

**Keywords:** ChatGPT, statistical analysis

## Abstract

Background: Content generated by artificial intelligence is sometimes not truthful. To date, there have been a number of medical studies related to the validity of ChatGPT’s responses; however, there is a lack of studies addressing various aspects of statistical analysis. The aim of this study was to assess the validity of the answers provided by ChatGPT in relation to statistical analysis, as well as to identify recommendations to be implemented in the future in connection with the results obtained. Methods: The study was divided into four parts and was based on the exemplary medical field of allergology. The first part consisted of asking ChatGPT 30 different questions related to statistical analysis. The next five questions included a request for ChatGPT to perform the relevant statistical analyses, and another five requested ChatGPT to indicate which statistical test should be applied to articles accepted for publication in *Allergy*. The final part of the survey involved asking ChatGPT the same statistical question three times. Results: Out of the 40 general questions asked that related to broad statistical analysis, ChatGPT did not fully answer half of them. Assumptions necessary for the application of specific statistical tests were not included. ChatGPT also gave completely divergent answers to one question about which test should be used. Conclusion: The answers provided by ChatGPT to various statistical questions may give rise to the use of inappropriate statistical tests and, consequently, the subsequent misinterpretation of the research results obtained. Questions asked in this regard need to be framed more precisely.

## 1. Introduction

At the end of July 2023, guidance related to the use of artificial intelligence, language models and chatbots was published by JAMA Network. The authors indicated that non-human artificial intelligence tools should not be credited as co-authors. The use of such tools should be transparently reported when writing manuscripts and other content [1]. A similar position has been expressed by the World Association of Medical Editors (WAME) [2]. An article published in June 2023 in *Allergy* pointed out that ChatGPT can generate reliable but incorrect and meaningless responses. For this reason, it is essential to keep expert supervision in mind when publishing this type of material [3]. A number of articles have been published in recent months assessing the accuracy of ChatGPT’s answers. In an example study, a team of three oncologists with extensive professional experience assessed the correctness of answers to 30 questions. In total, 14% of the answers were incorrect, and thus there was a high risk of receiving incorrect recommendations [4]. Another study investigated the correctness of answers to questions on a highly demanding postgraduate exam, namely the European Exam in Core Cardiology (EECC). The results indicated significant success of ChatGPT in the EECC; however, it cannot replace qualities that doctors should possess such as critical thinking, innovation and creativity [5]. An interesting study was also conducted on 15 different sub-disciplines of otolaryngology in which the dataset contained 2576 questions, i.e., 479 multiple-choice and 2097 single-choice. The percentage of multiple-choice questions answered correctly by ChatGPT was lower than that for single-choice questions. The highest percentage of correct answers given was related to allergology, while the lowest was related to legal aspects of otolaryngology [5]. The provision of incorrect answers by ChatGPT with the occurrence of lower levels of statistical analysis carried out in recent years may further reduce the confidence of the public in science. In one of the most cited papers published in *PLOS Medicine*, Professor John P. A. Ioannidis argued that the results of the majority of scientific publications are fraudulent [6]. Other data published in *PLOS ONE* show that between 1998 and 2020, there was unfortunately no significant change in the frequency with which medical journal editors used statistical reviewers [7]. In addition, another factor contributing to the poorer quality of the analyses conducted is the non-practical way in which biostatistics is taught. The statistical recommendations published in 2016 by *PLOS Biology* point to the need for education through critical analysis of accepted manuscripts [8]. Such aspects mentioned above, combined with misinformation provided by ChatGPT related to the way statistical analysis is carried out, may unfortunately contribute to an even greater deterioration in the quality of published research results and thus the subsequent implementation of inappropriate recommendations. What is lacking in the literature is an analysis of the answers provided by ChatGPT about various aspects of statistical analysis. To the best of my current knowledge, this is the first study of its kind to be carried out. As *Allergy* is the official journal of the European Academy of Allergy and Clinical Immunology, the first aim of this research was to determine the validity of the statistical analyses used by the authors of the articles published in this journal. The second objective was to analyse the validity of the statistical analyses carried out by ChatGPT on the datasets created, the subjects of which were related to allergology. The third objective of this research was to investigate the correctness of ChatGPT’s answers to questions on various statistical issues, using the medical field of allergology as an example. The fourth objective of the manuscript was, using one statistical question asked of ChatGPT as an example, to indicate the variety of answers given. The final objective was to provide statistical recommendations to bear in mind when considering the use of ChatGPT for conducting statistical analyses.

## 2. Materials and Methods

### 2.1. ChatGPT Statistical Skills

Due to its wide and general availability, ChatGPT-3.5 was used in the present study. The study was divided into four parts. ChatGPT-4 was excluded from this study due to its cost and limited availability. The questions that ChatGPT answered were related to broad statistical analysis and were based on different groups of allergy patients. The first part of the survey consisted of asking ChatGPT 30 different questions about which statistical tests should be applied to particular cases. Other questions asked included the assumptions of the statistical tests necessary for their application, as well as other aspects related to the performance of the statistical analysis. The second part of the research carried out involved a request to ChatGPT for it to carry out relevant statistical analyses based on five specific datasets. The validity of the statistical tests used was assessed. The third part of the study involved five articles published in *Allergy*. ChatGPT provided answers to questions about what kind of statistical test it would apply to specific studies conducted by these authors, the results of which were subsequently published in *Allergy*. The articles were selected based on the various statistical tests applied. In order to show the variety of answers given, one question related to the conduct of a statistical analysis was selected and then asked several times of ChatGPT. Based on the answers given by ChatGPT to the 40 questions, the statistical recommendations mentioned in the introduction were created. The answers given were scored using a 0–1 system, where 0 was the wrong answer and 1 was the correct answer. Incorrect answers were related to the wrong choice of statistical tests, the tests’ assumptions, and other aspects related to conducting a statistical analysis, such as the correct description of the obtained test results, removal of outliers, etc. In an additional supplement file (Appendix A) to this manuscript, screenshots of the questions asked of ChatGPT are included, along with its answers.

### 2.2. Statistical Analysis

For each part of the survey, the percentage of correct and incorrect answers given was calculated. A chi-square test was used to check whether ChatGPT gave more correct or incorrect answers. Statistical analysis was conducted using the IBM SPSS Statistics 25 Package (IBM SPSS Statistics Armonk, NY, USA: IBM Corp.). A *p* value < 0.05 was taken as the level of statistical significance.

## 3. Results

### 3.1. Correctness of ChatGPT’s Answers to Questions Related to Statistical Analysis

The following table lists the 30 questions asked of ChatGPT related to statistical analysis in the broadest sense. The number of correct answers out of 30 possible questions was 18 (60%). Of the answers given, 12 were incorrect. The difference in the number of correct and incorrect answers given was not statistically significant, χ^2^(1) = 1.2; *p* = 0.27. However, the responses associated with the need to use non-parametric equivalents of statistical tests were noteworthy. This refers to ChatGPT’s lack of consideration of the scale of the analysed variables, for example, related to ordinal nature, low sample size and other assumptions necessary for the application of specific tests. The remaining wrong answers given were to do with the wrong choice of statistical test, the wrong measure of effect size and the wrong descriptive statistics to be included in the description of the results obtained (Table 1). For this reason, statistical questions need to be formulated more precisely, bearing in mind the assumptions necessary for the application of specific statistical tests (equality of groups compared, normality of distribution, etc.).

### 3.2. Correctness of ChatGPT’s Statistical Analyses for Five Specific Studies

In terms of the correctness of ChatGPT’s five different statistical analyses, as was observed for the 30 questions in Table 1, ChatGPT did not consider sample size or the scale on which the variable being analysed was measured. Failure to take this type of information into account also resulted in the incorrect indication of the appropriate measure of effect size to be used in a particular task (Table 2). In order to carry out a statistical analysis when asking ChatGPT questions involving concrete data, questions should be asked in a precise way. It is recommended to include information on the need to test the assumption of normality of distribution, assessment of sample size, homogeneity of variance, etc.

### 3.3. Addressing the Statistical Tests That Should Be Used in Published Articles in Allergy

ChatGPT gave correct answers to the questions related to three of the five selected articles published in *Allergy*. Once again, the incorrect responses were related to the fact that ChatGPT did not consider aspects such as sample size and the equality of the groups of people being compared, and therefore did not select the appropriate effect size measure (Table 3) [9,10,11,12,13]. The recommendations related to the following statements are the same as those made when describing Table 2 (test assumptions, measurement scale of variables, etc.). It is important to remember to specify the questions being asked.

### 3.4. Repeatability of the Answer Given to a Specific Statistical Problem

When asking ChatGPT questions related to different aspects of the statistical analyses being conducted, it was noted that there were discrepancies in the answers given to the same questions. One question was selected from the 30 questions presented in the first table and then presented to ChatGPT three times to show the discrepancies in the answers given. Of the 30 questions presented in Table 1, the second was selected for analysis due to the scale of the variable under analysis. The first answer given recommended the use of a repeated-measure analysis of variance, without indicating the validity of using the Friedman test in this particular case. The second answer given to the same question indicated a chi-square test, but the subsequent response acknowledged that this test is used for independent observations. For this reason, ChatGPT noted the validity of using the McNemar test in this particular case, which was also not an appropriate choice. A surprising answer was received the third time. When the question was asked a second time, the response obtained, despite mentioning the chi-square test, did not support the use of this test. When asked the same question a third time, ChatGPT indicated that “Since you have multiple measurements for each patient taken at different times, a suitable statistical test to use in this scenario is the Chi-squared test for independence.” Despite previously pointing out the unreasonableness of using a chi-square test, the subsequent answer given contradicted this earlier response. Scans of the responses provided by ChatGPT are included in a supplement file (Appendix A).

## 4. Discussion

### 4.1. ChatGPT’s Statistical Skills

The present research is, to the best of my knowledge, the first on ChatGPT’s statistical skills. Using the medical field of allergology as an example, including articles published in *Allergy*, a study was carried out to analyse the accuracy of ChatGPT’s answers to questions related to statistical analysis in a broad sense. Unfortunately, the results obtained indicate that out of 40 possible questions, only half were correctly answered, which had to do with the lack of specification of questions in which assumptions such as sample size, measurement scale of analysed variables, etc., had to be taken into account. A review article published in *Health and Technology* pointed out that the use of artificial intelligence tools can help to facilitate comprehensive statistical analysis, including troubleshooting missing data [14]. ChatGPT can play a significant role in conducting statistical analyses, although without human oversight of every stage of a piece of research, it is not possible to adequately report the results of the research [15]. A study on how AI-based transformers can help with the design of epidemiological studies also indicated, as this manuscript does, that ChatGPT can give methodologically flawed answers. The lowest scores were recorded for this section [16]. An interesting article was recently published in *Schizophrenia* with the following title: “ChatGPT: these are not hallucinations—they’re fabrications and falsifications”. Emsley R. pointed out that questions on general methodology produced general and down-to-earth answers. Difficulties were encountered in obtaining slightly more detailed answers related to a statistical analysis plan [17]. This confirms the results of the studies conducted in this work. At present, unfortunately, the quality of statistical reporting, interpretation and presentation of data in scientific articles is poor [18]. One recently published survey indicated that only 39% of accepted articles on various aspects of COVID-19 met the requirements of statistical validity [19]. An increasing number of journals are publishing statistical recommendations; however, it is difficult to know at this point whether such articles are improving reporting standards [20,21,22]. The research conducted in this work indicated that the main errors associated with the advice given related to the failure to consider a number of assumptions when selecting specific statistical tests. One of the main mistakes most frequently made in journals is the application of parametric statistical tests to non-parametric data [19,23]. The ordinal scale of the variables analysed, low size and unequal nature of the groups of individuals compared, and disorders of normality of distribution are all indications for the use of non-parametric equivalents of statistical tests [24,25]. Unfortunately, using the example of many questions related to the medical field of allergology, the advice provided by ChatGPT did not take such assumptions into account, which, with the low level of statistical analysis being performed today, may unfortunately exacerbate the problem instead of helping. According to the recommendations indicated in *PLOS Biology*, it is now generally recommended to provide practical instruction in performing statistical analyses, e.g., using examples of published articles in which specific statistical tests have been applied [8]. One published comparison of student performance in essay writing with or without the use of ChatGPT-3 indicated that the use of this type of tool did not play a major role in this task. This was related to an increased risk of plagiarism, a lack of originality and a lack of acceleration in essay writing [26]. An article published at the end of 2022 in *Nature* pointed out researchers’ concerns about students writing homework with the help of artificial intelligence [27]. The non-practical way in which statistical analysis is taught today, coupled with ChatGPT’s misguided advice on the subject, may unfortunately result in the implementation of inappropriate solutions based on published articles in the future. The responses obtained from ChatGPT also indicated incorrect measures of effect size in specific commands. Among other things, ChatGPT did not take into account the number of levels at which a particular variable was measured, nor its scale, and these are among the factors that play a role in the selection of an appropriate parameter [28]. ChatGPT’s indication of an inappropriate measure of effect size may also be a cause of misinterpretation of the research results obtained. It is noteworthy that, in the survey conducted, ChatGPT gave divergent answers to the same question related to which statistical test should be used. This made it all the more surprising to see that ChatGPT, after initially analysing the lack of validity of using a particular statistical test, indicated the same test about which it had expressed the opposite opinion a moment earlier, when presented with the question a second time. This confirms information published previously indicating that ChatGPT displays different answers when repeatedly presented with the same prompt [29]. Asking the same question may result in different answers [30]. A command such as “regenerate answer” may lead to different conclusions [31]. The provision of different answers by ChatGPT, combined with inappropriate advice related to the conduct of statistical analysis, may unfortunately result in a decrease in the public’s confidence in science. The results of a survey of adults in the United States indicated that over-reliance on ChatGPT can contribute to the spread of misinformation and subsequent risks to our health. This risk may be related to over-reliance on chatbots [32]. This manuscript is about the statistical skills of ChatGPT. However, alternative artificial intelligence tools such as BARD by Google are recommended for the future. A comparison with other artificial intelligence models would allow the reliability of the research to be assessed. In other words, the use of multiple artificial intelligence tools to answer the same questions related to conducting a statistical analysis would illustrate the existing consistency and variability in the recommendations given.

### 4.2. Statistical Recommendations Related to the Use of ChatGPT

Based on the research conducted, statistical recommendations related to the use of ChatGPT are indicated below. An initial recommendation for authors wishing to use ChatGPT’s indicated answers to questions related to statistical analysis in the broadest sense concerns the initial analysis of the assumptions necessary to apply specific tests. Before the authors decide, for example, to apply a statistical test indicated by ChatGPT, they should first examine the individual assumptions. Assumptions such as normality of distribution, group size, equality of groups, the scale on which the variable being analysed is measured, etc., must be taken into account. ChatGPT should not be asked general questions related to what kind of analysis should be applied, and the highlighted assumptions should be considered. In other words, it is advisable to ask more precise questions about whether 16 samples are enough to apply an analysis, or, for example, whether the measurement scale on which a variable is measured allows the application of a particular test. Without a specific question, the percentage of statistical advice provided by ChatGPT that is appropriate may be low. The second recommendation is to ask the same ChatGPT question several times because, as indicated in this manuscript, there is the possibility that it will provide quite divergent answers and thus subsequently suggest the wrong advice. Once different answers to a single statistical question have been obtained, authors should analyse these in depth, choose the most optimal answer to the best of their knowledge and then, if there are further doubts, relate the chosen answer to the relevant references. One option is also to refer to published statistical recommendations in medical journals such as those published by this journal, among others [33]. This is another such recommendation. In these times of developing artificial intelligence, it is all the more advisable to include statistical guidance on the websites of medical journals, or for recommendations to be published in said journals. Before authors submit a manuscript to a particular journal, they should first familiarise themselves with these recommendations. It should also be noted that authors who have used ChatGPT may not specifically have included this information in the submitted manuscript. A review by the authors of guidance obtained in this way should take place before submission of a manuscript and be indicated in the cover letter. The next recommendation is to forward to an expert in biostatistics any statistical analyses carried out for which the authors have applied advice from ChatGPT. Information about the statistical review carried out by this expert should also be included in the cover letter. The last recommendation has to do with presenting at major scientific conferences/congresses on the reliability of the information provided by ChatGPT related to statistical analysis in the broadest sense. These include major events attended by members of the International Committee of Medical Journal Editors and the World Association of Medical Editors. Making the scientific community aware of the most common possible errors in statistical advice provided by ChatGPT could contribute to even greater attention being paid to the aspects discussed in this manuscript.

## 5. Conclusions

At a time when there is a lower level of statistical analysis being conducted, we should have limited confidence in the advice provided by ChatGPT related to conducting statistical analyses.

## Figures and Tables

**Table 1 healthcare-11-02554-t001:** Correctness of ChatGPT’s answers to 30 different questions related to statistical analysis.

Question	Comment
1. I would like to test whether there is a statistically significant association between serum IL-13 levels and IgE levels in 16 patients with bronchial asthma. Which statistical test should I use?	Selection of an incorrect correlation coefficient (low sample size). It is advisable to ask a more precise question, such as whether, for example, the indicated sample size is sufficient to apply the indicated test.
2. I would like to test whether there are statistically significant differences in cough severity in the group of 50 asthmatic patients I studied. Measurements were taken five times during treatment. Each patient could tick a response of strong, moderate, or weak cough severity. Which statistical test should I use?	Failure to indicate the need for a non-parametric equivalent of a statistical test (ordinal nature of the variable). When asking questions, it is advisable to point to the measurement scale on which the variable is measured.
3. I would like to predict by regression analysis whether the severity of the disease as measured by the quality of life questionnaire can be predicted from the number of eosinophils and age. Can I carry out such an analysis?	Failure to indicate a possible correlation between the analysed predictors. When asking questions, it is important to keep in mind the assumptions of statistical tests, including regression analysis.
4. Using the Mann-Whitney U test, I found that there were statistically significant differences between the Stevens-Johnson syndrome group and the control group in terms of IL-13 levels. I was pleased to write that the results obtained indicated that the mean IL-13 levels in the Stevens-Johnson syndrome patients appeared to be statistically significantly higher compared to the control group (*p* < 0.001). The mean level of IL-13 in the Stevens-Johnson syndrome patients was 274.6 ± 493.0 pg/mL, the range in healthy subjects was 3.1 ± 0.1 pg/mL. Have I described the research results obtained well enough?	Failure to indicate a recommendation related to the need to take into account other descriptive statistics such as the median. It is advisable to ask more precise questions in the style of, for example, “Are descriptive statistics such as mean and standard deviation sufficient for the statistical test used?”
5. By analysis of variance, I found that there were statistically significant differences between the control group, patients with mild and severe COVID-19 in terms of eosinophil levels. One group comprised 30 patients, the second 12 and the third 20. The variances in the comparison groups are not homogeneous. Which post-hoc test should I use?	Selection of an appropriate post hoc test.
6. In my analysis of variance with repeated measures, I wanted to test whether individual measurements of IFN-γ levels in patients with allergic rhinitis change over time. I obtained a significance level for the Mauchley test of 0.02. Do I understand correctly that the assumption of sphericity of variance is met and I do not need to apply the corresponding correction?	Correct interpretation of the indicated assumption.
7. I want to carry out a cluster analysis using the k-means method. I want to distinguish 2 distinct clusters of my study group of asthmatic patients on the basis of haemoglobin (mg/dL) and interleukin (pg/mL) levels. Can I automatically use these two variables in this type of analysis?	Indication of the need for standardisation of variables.
8. I would like to compare several groups of patients with eosinophilia in terms of another blood morphology parameter. Can I use the *t*-student test several times for this purpose?	Indication of the error that would be conducting a series of Student’s *t*-tests.
9. I would like to test the assumption of normality of the distribution of the variable I am analysing. A total of 78 people with contact eczema took part in the study. Can I use the Kolmogorov-Smirnov test for this purpose?	Indication of the appropriate statistical test for this sample size.
10. The reviewer recommended that I analyse the interaction study group * gender, rather than just examining the asthma patient group as a factor. Can I therefore test separately in one group and separately in the other group using the Student’s *t*-test to see if there are differences between women and men?	Failure to carry out simple main effects analysis and separate Student’s *t*-tests. It is advisable to ask a more precise question along the lines of, for example, “How can I analyse the interaction of two independent variables?” Assumptions such as sample size, measurement scale of the variable, etc., should also be indicated.
11. I found a statistically significant relationship between the pass rate of the allergy exam (yes or no) with the gender of the medical students studied (Female/Male). I would like to explore the strength of this relationship. Can I use Cramer’s V-factor for this purpose?	Indication of wrong measure of effect size (instead of *Phi φ*). It is advisable to indicate the measurement scale of the variable and whether the variable is multicategorical or, for example, bicategorical.
12. Can I use the Sobel test to determine the significance of an indirect relationship in a mediation analysis for a sample of less than 50 allergy patients?	Appropriate guidance relating to mediation analysis with this sample size.
13. I would like to check whether there is a statistically significant relationship between the severity of atopic dermatitis symptoms (minor/moderate/significant) and Beck Depression Scale scores. Can I use Pearson’s correlation coefficient for this purpose?	Selection of the wrong correlation coefficient (not taking into account the scale of the variable). It is recommended to indicate the measurement scale of the variables analysed.
14. I am in the process of testing the assumptions of the regression analysis. I would like to try to predict the quality of life level of patients with severe COVID-19 based on eosinophil levels. The Durbin-Watson test statistic came out equal to 0.9. My interpretation is that the assumption of correlated residuals is not broken, i.e., the residuals are not correlated. Am I interpreting the result correctly?	Correct interpretation of the Durbin–Watson test.
15. I am carrying out a one-way analysis of variance to compare three groups of people allergic to nuts divided according to the intensity of symptoms experienced. The dependent variable is the SDQ-7 sleep disorders questionnaire score. The difference between the variances in the compared groups is statistically significant. Do I understand correctly that I do not need to apply the Welch or Brown-Forsythe correction? +	Correct indication related to the need to apply the Brown–Forsythe correction
16. I would like to look for statistically significant predictors of quality of life in asthma patients. The group size is 100 patients. Can I perform a regression analysis by including twenty predictors?	Appropriate indication of too few observations for this number of predictors.
17. I carry out a logistic regression analysis to investigate the impact of different patient characteristics on the incidence of asthma. In my study, the observations come from repeated measurements. Can I carry out this type of analysis?	Proper indication of the analyses that could be carried out.
18. In the article I want to send to Allergy I have included a description of the statistical tests used. I wrote that in this manuscript I used the Student’s *t*-test for independent samples, the Kruskal-Wallis test and Pearson’s correlation analysis. I have also included a sentence about the fact that I have used descriptive statistics like mean and standard deviation. Is such a description sufficient?	Appropriate advice on the description of the analyses carried out.
19. I observed that there were statistically significant differences between patients with food allergy and the control group with regard to the levels of the interleukin I studied. The *p*-value was 0.049. This shows that the research results obtained are significant, it is even a scientific discovery. No one has done such research until now. The *p*-value obtained means, in your opinion, that the results I have obtained play a significant role?	Appropriate advice on the application of the effect size measure.
20. I compare 6 groups of patients with atopic dermatitis in terms of IL-25 levels. Among many existing post-hoc tests, can I choose one that I think will increase my chance of obtaining statistically significant differences? In doing so, I want to increase my chances of being accepted in the Allergy article. I will ask for advice.	Appropriate indication of ethical issues.
21. I would like to find out whether in my study group of women and men with allergic conjunctivitis, the treatment applied improves their quality of life. I took measurements several times during treatment. Which statistical test should I use?	Failure to demonstrate the validity of conducting an analysis of variance in a mixed design. In addition to the most important assumptions of statistical tests, it is advisable to indicate whether each factor is to be analysed separately or whether the author is interested in, for example, the interaction between within-subject and between-subject factors.
22. The reviewer of the article I sent to Allergy recommended analysing the outlier cases. To this end, I used the Cook distance. The value obtained was 1.2. I remember from statistics class that this means that the case is not an outlier. Do I remember correctly?	Failure to indicate a slight overshoot of Cook’s distance and therefore the possible impact of outlier cases. For more advanced statistical tests, it is recommended to check the recommendations made by ChatGPT against the relevant statistical references.
23. I have been studying the severity of depressive symptoms in a group of patients with allergies for several years. I aim to assess the impact of the psychotherapy I use on the subjects’ Beck Depression Scale scores before and after therapy. Twenty-five took part in the study at the start of treatment, while 15 completed it. Can I use the *t*-student test for dependent samples in this case?	Failure to indicate the need for a non-parametric equivalent of a statistical test (low sample size). In addition to assessing the sample size, it is advisable to check that the groups being compared are equal.
24. A reviewer in Allergy recommended that I check whether the use of the drug I indicated in the manuscript has a positive effect on the treatment of atopic dermatitis. Before and after therapy, I checked whether specific symptoms were present (yes or no). In order to check whether the use of the drug causes the disappearance of symptoms, can I use the Wilcoxon test?	Failure to consider the scale of the variable being analysed and thus to indicate the inappropriate choice of statistical test (Wilcoxon instead of McNemar). It is recommended that the measurement scale of the variables being analysed is carefully defined.
25. Analysis with the Kruskal-Wallis test indicated that there were statistically significant differences between the 3 groups of patients with photodermatosis I compared. How do I find out exactly which groups differ in a statistically significant way?	Indication of an appropriate post hoc test for a given statistical test.
26. I would like to investigate the effect of several collinear independent variables on survival time. I remember from a biostatistics class that Cox regression is used for such a purpose. Is the right test going to be used?	Relevant advice on Cox regression.
27. The resulting *p*-value for Levene’s test of homogeneity of variance was 0.02. Do I understand correctly that in the *t*-student test I use for independent samples, I do not need to read the test value with correction for heterogeneous variances?	Proper interpretation of the Levene test.
28. I would like to investigate the strength of the association between bronchial asthma patients’ Beck Depression Scale scores (max 40 points) and the form of nutrition (yes or no) they use. How do I explore the strength of the relationship? Could this be a phi statistic?	Indication of the wrong measure of effect size, i.e., phi instead of eta. When choosing an appropriate measure of effect size, it is advisable, among other considerations, to indicate the scale of measurement and how many categories are used to examine the variables being analysed.
29. I want to get the maximum effect size. Should I include as many patients with allergic rhinitis as I can in the study?	Appropriate advice on the measure of effect size.
30. I received from the reviewer of an article sent to Allergy that I should check the effect of outlier cases on the result of the correlation analysis I used. Can I simply delete this result?	Appropriate advice related to the removal of outlier cases.

Note: “*”: Interaction of two variables, the symbol denotes interactions between variables.

**Table 2 healthcare-11-02554-t002:** Correctness of ChatGPT’s statistical analyses for five specific studies.

Statistical Analysis	Comment
1. I would like to check if there is a statistically significant association between IgE and IL-18 levels. The results obtained for IgE levels (IU/mL) in a group of 12 asthmatic subjects are: 896, 467, 890, 765, 490, 589, 201, 875, 743, 910, 772, 498. For IL-18 levels (pg/mL), the results obtained for the same in order as before for the subjects are: 543, 323, 652, 423, 456, 499, 342, 290, 499, 502, 399, 390. Can you check in Python whether there is a statistically significant relationship between the analysed variables? I want you to calculate the correlation coefficient and the *p*-value. -	Selection of an incorrect correlation coefficient (low sample size).
2. In a group of 15 food-allergic patients, I would like to test whether there are statistically significant differences in cough severity (weak/moderate/significant). This was measured three times. The results obtained for the first measurement in order are: 3, 2, 3, 2, 2, 2, 3, 3, 3, 3, 2, 2, 2, 3, 3, 2. Subsequent results for these individuals are as follows: 2, 2, 2, 2, 2, 3, 2, 2, 2, 3, 1, 3, 2, 2, 3, 2. For the third measurement in the same order of persons the results were as follows: 1, 1, 1, 2, 2, 2, 1, 1, 1, 2, 2, 2, 3, 2, 1, 1. Could you apply a suitable statistical test and check if there are statistically significant differences between the three measurements? -	Failure to consider the scale of the variable being analysed and, therefore, to identify an inappropriate statistical test.
3. I carry out an analysis of the relationship between the gender of people with atopic dermatitis and the therapeutic effect of the dupilumab used. The first 15 subjects were female and the next 15 were male. Obtaining a therapeutic effect was marked as no or yes. In Women, the results were as follows: 0, 0, 0, 0, 0, 0, 1, 1, 1, 1, 1, 1, 1, 1, 0, 0 and 1. In Men, the results were as follows: 0, 0, 0, 0, 0, 0, 0, 0, 0, 0, 1, 0, 0, 0, 0, 0. I observed the presence of a statistically significant relationship, which indicates that in a higher proportion of Women compared to Men the drug worked. However, I would still like to investigate the corresponding effect size measure. Could you please calculate it for me? -	Indication of the wrong measure of effect size.
4. I am conducting a study to see if there are statistically significant differences between three groups of patients with asthma, namely divided by the severity of the characteristic symptoms of asthma. The dependent variable is the life satisfaction score. The first group includes 20 patients, the second group 10 and the third group 25. The scores obtained for the first group in order are: 13, 15, 16, 11, 15, 17, 13, 16, 18, 12, 11, 15, 18, 19, 20, 17, 17, 12, 11 and 15. For the second group of patients, the scores are as follows: 17, 18, 16, 19, 21, 14, 15, 16, 17, 21, while for the third group of patients: 23, 24, 19, 21, 20, 15, 15, 25, 26, 21, 17, 19, 22, 17, 18, 19, 23, 26, 21, 22, 19, 19, 20, 16 and 25 scores. Could you check if there are statistically significant differences between the groups I am comparing? -	Disregard of the unevenness of the groups of people being compared, as well as the size of the sample, and therefore selection of the wrong statistical test.
5. I investigated Il-21 [pg/mL] levels in a group of patients with atopic dermatitis before and after the implementation of drug therapy. Twenty patients took part in the study. The results for the first measurement are as follows: 890, 344, 259, 685, 456, 333, 289, 910, 899, 543, 467, 211, 453, 678, 211, 987, 878, 234, 367 and 459, and for the second measurement: 490, 290, 232, 451, 395, 322, 299, 432, 765, 322, 201, 123, 209, 444, 100, 333, 789, 178, 211 and 400. I used the *t*-student test for dependent samples, which contributed to the fact that I found statistically significant differences between the two time measurements. The results for the second measurement were found to be statistically significant lower. Did I do the analysis well and could you do it again for me? I want to find out if there was a statistically significant change in the second time period.	Failure to consider disturbances in the normality of the distribution and the low sample size, and thus selection of the wrong statistical test.

**Table 3 healthcare-11-02554-t003:** Correctness of ChatGPT’s indication of answers to statistical questions related to articles accepted for publication in *Allergy*.

Question	Comment
1. In an article published in July 2022 in Allergy, the authors compared 3 subgroups of patients divided by fractional exhaled nitric oxide (FeNO). Fifty-one people were included in one group, 46 in the second and 18 in the third. Among other things, the authors made comparisons in terms of eosinophilia counts. What statistical test do you think they could have used to check whether the three groups differed in a statistically significant way? [9]	Disregard of the unevenness of the groups and the low size of one of the groups, and therefore selection of the wrong statistical test.
2. In an article published January 2022 in Allergy, the authors investigated the effect of dupilumab on improving health-related quality of life. This was a phase 3 study of a certain project. The group of patients taking dupilumab included 438 patients, while placebo included 286. The authors observed the presence of statistically significant differences between the groups. Quality of life was assessed using a 22-item sinus-nasal test score. The maximum possible score was 110 points. What measure of effect size do you think they could have used to show the practical significance of the results obtained? [10]	Indication of the wrong measure of effect size to use.
3. In a paper published in January 2023 in Allergy, the authors wanted to assess the relationship between IgE levels for peach components and IgE levels for food, inhalant allergens and latex if there were at least 10 patients. What analysis for this type of sample size should they use? [11]	Indication of an appropriate correlation analysis.
4. Published in Allergy in December 2019, the results of the study looked at dual blockade of IL-4 and IL-13 with dupilumab. The authors used a one-way analysis of variance. They mentioned a comparison of standard deviations in independent groups. What F statistics correction should they apply when it would appear that the variances are not homogeneous? [12]	Indication of the appropriate adjustment to be applied.
5. In a letter to the editor published in February 2023 in Allergy, the authors investigated the early increase in serum specific IgG2 after allergen immunotherapy. Measurements were taken before the start of therapy, after 4 months and after 1 year. The group of patients treated with the 300-IR HDM tablet was divided into two subgroups of 25 patients in each group. What test should the authors apply to check for statistically significant differences in each of these subgroups between the three time periods being compared? [13]	Indication of an inappropriate statistical test due to not taking into account the size of the study group.

## Data Availability

Not applicable.

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
