# Peer review of "ChatGPT’s Skills in Statistical Analysis Using the Example of Allergology: Do We Have Reason for Concern?"

_healthcare, 2023, doi:10.3390/healthcare11182554_

Round 1

Reviewer 1 Report

The authors seek to evaluate ChatGPT's capabilities in performing statistical analyses and also it's ability to suggest methods in specific conditions as per their thirty questions. 

The paper is written well and is easy to follow through. However, there are some major concerns in the study. 

The authors ask ChatGPT for it's input on statistical analysis for the case of 16 samples in the list of thirty questions and also seek chatgpt to perform statistical analyses on 12 samples and note that the response was negative because chatgpt did not warn about sample size. This is however not sound. A generative model such as Chatgpt must be clearly asked about statistical test and also if the sample size is okay in order to generate an appropriate response. For instance when I copied the author's question and added "Also is 16 samples sufficient for the analyses" chatgpt responded that the sample size is considered low. Here is chatgpt's response when I asked the question that authors asked, but added the extra sentence:

Question:

would like to test whether there is a statistically significant association between serum IL-13 levels and IgE levels in 16 patients with bronchial asthma. Which statistical test should I use? Also is 16 samples sufficient for the analyses

Answer:

With a sample size of 16, it's quite small. Here are the implications:

  1. Power of the test: A small sample size reduces the power of your statistical test. This means you might not detect a significant correlation even if one exists (Type II error).
  2. Confidence Interval Width: The confidence intervals for the correlation will be wider, which means more uncertainty about the true population correlation.
  3. Risk of Outliers: Small sample sizes are more susceptible to the effects of outliers. A single unusual data point can significantly influence the correlation coefficient.

To determine if 16 samples are sufficient, you can conduct a power analysis. This will tell you the probability of detecting an effect, given your sample size, the anticipated effect size, and the desired significance level.

In practical terms, while 16 samples might provide preliminary insights, it's generally considered small in many clinical studies. For robust conclusions, especially in a variable field like medical research, larger samples are often preferred.

If you proceed with the analysis using the 16 patients, you should be cautious in interpreting the results. Even if you find a statistically significant correlation, the clinical significance and generalizability of the finding will be limited due to the small sample size.

To test for a statistically significant association between two continuous variables (serum IL-13 levels and IgE levels), you would typically use the Pearson correlation coefficient. This test measures the strength and direction of the linear relationship between the two variables.

However, before running the test, it's essential to check for the assumptions:

  1. Linearity: Both variables should have a linear relationship.
  2. Normality: Both variables should be approximately normally distributed.

If the assumptions are not met, you might consider the Spearman rank correlation as a non-parametric alternative.

A similar theme was found for other questions posed by the authors. When I pasted the second question from the table, chatgpt prompted that severity is ordinal and it suggested ordinal data based method, however authors note otherwise. 

I suggest authors to be precise in asking questions. In addition, it is understandable that chatgpt might not respond the same for the very same question and it might generate alternate responses, hence it is not effective to state that chatgpt definitively gives a wrong answer or a right answer. It can shuttle between right and wrong in a certain context. The authors rightfully discuss this in their methods and hence to deem a response negative or positive seems not straightforward, however authors can restructure the evaluation suggesting the answer could have used improvement from chatgpt due to reason X, Y etc.  

The authors have written the manuscript well and no concerns on quality of language. 

Author Response

Dear Reviewer nr 1,

Concern 1: “The authors seek to evaluate ChatGPT's capabilities in performing statistical analyses and also it's ability to suggest methods in specific conditions as per their thirty questions. The paper is written well and is easy to follow through. However, there are some major concerns in the study. The authors ask ChatGPT for it's input on statistical analysis for the case of 16 samples in the list of thirty questions and also seek chatgpt to perform statistical analyses on 12 samples and note that the response was negative because chatgpt did not warn about sample size. This is however not sound. A generative model such as Chatgpt must be clearly asked about statistical test and also if the sample size is okay in order to generate an appropriate response. For instance when I copied the author's question and added "Also is 16 samples sufficient for the analyses" chatgpt responded that the sample size is considered low. A similar theme was found for other questions posed by the authors. When I pasted the second question from the table, chatgpt prompted that severity is ordinal and it suggested ordinal data based method, however authors note otherwise. I suggest authors to be precise in asking questions. In addition, it is understandable that chatgpt might not respond the same for the very same question and it might generate alternate responses, hence it is not effective to state that chatgpt definitively gives a wrong answer or a right answer. It can shuttle between right and wrong in a certain context. The authors rightfully discuss this in their methods and hence to deem a response negative or positive seems not straightforward, however authors can restructure the evaluation suggesting the answer could have used improvement from chatgpt due to reason X, Y etc.”

Answer 1: I would like to thank the reviewer for his positive opinion and for pointing out a critical comment to correct the written manuscript. Based on advice from the second reviewer, the revision of the manuscript consisted of expanding the discussion and limitations of the written manuscript. The effectiveness of determining whether ChatGPT gives the wrong or correct answer may indeed be somewhat questionable. For this reason, in line with the advice received from the reviewer, the column indicating whether ChatGPT gave the correct answer or not was removed from the individual tables. The recommendations written earlier additionally pointed out this type of aspect to look out for when asking ChatGPT statistical questions. In addition, in the questions included in Table 1 for which an unsatisfactory answer was obtained, in addition to the removal of the central column, the comments indicated next to it have been expanded to include the recommended advice. For Tables 2 and 3, general recommendations related to the answers provided are indicated next to their description. Sentences indicating the percentage of correct answers given have been corrected accordingly.

Reviewer 2 Report

The paper presents an interesting exploration into the capacity of ChatGPT to provide accurate statistical analysis recommendations within the domain of allergology. The study's objectives and methodology are well-outlined in the abstract, with the paper delving into the challenges posed by AI-generated content's veracity. However, upon closer examination, several concerns come to light that warrant addressing before publication.

Concern 1: Single-Person Conducted Study and Assessment

The primary concern lies in the sole involvement of a single individual in conducting the entire study, including determining ChatGPT's response correctness. This approach raises potential issues of personal bias and subjectivity influencing the assessment of response validity. A more rigorous evaluation could be achieved through independent validation by multiple researchers or experts in the field, thus enhancing the study's objectivity and reliability.

Concern 2: Lack of Exploration of Fine-Tuning Potential

While the study raises intriguing questions regarding ChatGPT's statistical analysis capabilities, the absence of discussion around the potential of fine-tuning the model for specific tasks is noteworthy. The authors could provide insights into the feasibility and benefits of adapting ChatGPT or other large language models (LLMs) for precise medical statistical analysis. Exploring the use of fine-tuned models in contrast to the base ChatGPT could provide a more comprehensive understanding of their potential in this context.

Concern 3: Comparative Analysis with Alternative Tools

The study focuses solely on ChatGPT's performance in medical statistical analysis, overlooking the presence of alternative AI tools such as BARD by Google. To provide a holistic perspective on LLM behavior, it would be valuable to include a comparison with other AI models, thereby enhancing the study's credibility. Employing multiple AI tools to address the same questions would better elucidate the consistency and variability in recommendations, contributing to a more comprehensive field assessment.

The overall writing is acceptable; however, some minor checks are needed.

Author Response

Dear Reviewer nr 2, 

“Concern 1: Single-Person Conducted Study and Assessment

The primary concern lies in the sole involvement of a single individual in conducting the entire study, including determining ChatGPT's response correctness. This approach raises potential issues of personal bias and subjectivity influencing the assessment of response validity. A more rigorous evaluation could be achieved through independent validation by multiple researchers or experts in the field, thus enhancing the study's objectivity and reliability.”

Answer 1: Thank you for your valuable advice. As a single author of a manuscript, I have many years of experience as a reviewer and statistical editor for numerous journals. I am an ISI (International Statistical Institute) Elected Member. I also review for statistical validity the Global Burden of Disease Study articles submitted by the authors. This type and all other statistical activity allowed me to carry out the present research and, consequently, to write the manuscript. In future, on the basis of the feedback received from the reviewer, an additional review will be carried out by an appropriate expert co-author before the manuscript is sent. In line with the advice received in the second review received, additional changes were made to the manuscript, including the deletion of one column in which the correctness of the answers given was indicated.

“Concern 2: Lack of Exploration of Fine-Tuning Potential

While the study raises intriguing questions regarding ChatGPT's statistical analysis capabilities, the absence of discussion around the potential of fine-tuning the model for specific tasks is noteworthy. The authors could provide insights into the feasibility and benefits of adapting ChatGPT or other large language models (LLMs) for precise medical statistical analysis. Exploring the use of fine-tuned models in contrast to the base ChatGPT could provide a more comprehensive understanding of their potential in this context.”

Answer 2: Thank you for another valuable piece of advice obtained. In line with the advice received, a few relevant sentences on the subject were included in the discussion.

“Concern 3: Comparative Analysis with Alternative Tools

The study focuses solely on ChatGPT's performance in medical statistical analysis, overlooking the presence of alternative AI tools such as BARD by Google. To provide a holistic perspective on LLM behavior, it would be valuable to include a comparison with other AI models, thereby enhancing the study's credibility. Employing multiple AI tools to address the same questions would better elucidate the consistency and variability in recommendations, contributing to a more comprehensive field assessment.”

Answer 3: I thank the reviewer also for and this kind of valuable recommendation pointing out another limitation of the study I performed. The topic of the article is about ChatGPT's skills, however, other artificial intelligence tools should indeed be referred to in the future. This is a very good point. For this reason, the limitations of the written manuscript in which this aspect is briefly discussed have been extended.

Round 2

Reviewer 1 Report

The authors have revised the manuscript appropriately and have stated clearly how the questions were framed and how they should be framed and what are the deficiencies in the answer. This version presents clear readability and is acceptable. 

No major concerns on quality of language. 

Author Response

Dear Reviewer nr 1,

Thank you for the positive feedback and for accepting the revised version of the manuscript.